# An Insight into *Cuscuta campestris* as a Medicinal Plant: Phytochemical Variation of *Cuscuta campestris* with Various Host Plants

Dariush Ramezan [1,*], Yusuf Farrokhzad [2], Meisam Zargar [3,*], Gani Stybayev [4], Gulden Kipshakbayeva [4] and Aliya Baitelenova [4]

1 Department of Horticulture and Landscaping, Faculty of Agriculture, University of Zabol, Zabol 98615-538, Iran
2 Department of Horticultural Science, Faculty of Agriculture, Tarbiat Modares University, Tehran 14115-336, Iran
3 Department of Agrobiotechnology, Institute of Agriculture, RUDN University, 117198 Moscow, Russia
4 Department of Plant Production, Faculty of Agronomy, S. Seifullin Kazakh Agrotechnical University, Astana 010000, Kazakhstan
* Correspondence: drhorticul@uoz.ac.ir (D.R.); zargar_m@pfur.ru (M.Z.)

**Abstract:** *Cuscuta campestris* is a holoparasitic plant that depends on the host for water, nutrients, and photosynthetic substances. The purpose of this research was to evaluate the effects of the host species on the content of bioactive and health-promoting substances in the *Cuscuta* seeds to test the following hypothesis: these substances are more induced if the hosts are herbs. The studied hosts were herbs (thyme, basil, and onion) and non-herbs (alfalfa and tomato). The results showed that the carotenoid accumulation in seeds developed on basil and thyme was the maximum. The extracts of seeds grown on thyme and onion had significantly more galactitol, total polysaccharide, and antioxidant activity than other hosts. Quercetin, kaempferol, and total flavonoids were higher in the seeds set on onion. The highest content of bergenin was recorded on thyme with no significant difference with onion. The extract of seeds grown on thyme had more total phenolics, followed by tomato and basil. Analysis of the phytosterol content of the seeds showed that campesterol was the minimum in seeds grown on alfalfa and stigmasterol was lowest in seeds grown on tomato than other hosts. Additionally, $\beta$-sitosterol increased in seeds developed on basil, onion, and thyme, and $\Delta^7$-avenasterol increased in seeds set on thyme and basil. Overall, the content of total phytosterols was higher in seeds developed on basil, onion, and thyme. The results were suggestive of the proper health-promoting levels of dodder seeds developed on medicinal plants such as onion and thyme for pharmaceutical and food applications.

**Keywords:** bergenin (cuscutin); DPPH radical scavenging; flavonoid compositions; galactitol (dulcitol); phytosterols; total polysaccharides

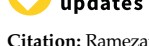



## 1. Introduction

The plants of the *Cuscuta* genus belong to the Cuscutaceae family and include about 200 species, all of that survive as stem holoparasites on host plants. The plants of the *Cuscuta* genus lack roots and well-expanded leaves, and their vegetative part is only a stem. The dodder parasite encircles the plants and entry into the host foliage through haustoria and makes direct junctions with the host vascular bundles to elicit water, carbohydrates, and other elements [1]. Parasitic plants of the *Cuscuta* genus have either no or only small amounts of chlorophyll and regularly absent photosynthetic activity [2]. Therefore, plants of this genus import all nutrients, photosynthetic assimilates, macro- and micronutrients, water, plant growth regulators, and other bioactive compounds from their host plant [3]. Additionally, recent reports have shown that RNAs move reciprocally between the host and the parasite through the parasite–host connections of the dodder plant with its hosts and affect a large number of genes [4].

Field dodder (*Cuscuta campestris* Yunck.) is the most common species among approximately 200 species of the *Cuscuta* genus in terms of distribution in different habitats worldwide [5]. Dodder has a wide host range and worldwide distribution. Despite high agronomical damages induced by *Cuscuta*, our knowledge of the associations between *Cuscuta* and its hosts is relatively limited compared to our knowledge of pathogenic fungi, bacteria, and viruses [6]. Environmental conditions (abiotic stresses such as salinity) [7], the growth stage of the host plant (cell wall lignification) [8], and host species affect the amount of biomass and biochemical quality of the dodder plants. In addition, several studies indicated that holoparasitic species have self-control mechanisms on their own metabolites, since their metabolic contents differ substantially from their host biomass [9,10]. However, Kumar and Amir [11] revealed that the type of host significantly changes the primary metabolism in the holoparasitic plant, and due to similar patterns in the profile of these metabolites in different organs, there is also a kind of self-regulation.

*Cuscuta* has a high reputation as a medicinal plant in traditional Chinese medicine. This plant has many medicinal properties [12]. The seed of this plant is one of the most common herbal compounds used to improve liver and kidney function and other cases in traditional Chinese medicine [13]. Dodder seeds are used as a traditional medicine to strengthen sexual power and function of the reproductive system, prevent and treat cardiovascular diseases, treat osteoporosis, and prevent aging [14]. In Iranian ethnobotany, the decoction of the seeds of this plant is used to cure people with sexual impotence, and it is recommended that medicinal plants should be the host of the dodder to take full advantage of its properties. It has been reported that most species of the *Cuscuta* genus have water-soluble phenolic substances, such as hyperoside, quercetin, astragalin, camphor, etc. [15]. Kaempferol, quercetin, hyperoside, astragalin, and lignans can be mentioned among the flavonoid compounds found in the genus *Cuscuta*, which play a significant role in the therapeutic properties of this plant against various diseases [16].

Sweet basil (*Ocimum basilicum*) is a significant commercial herb that is commonly cultured throughout the world. Basil essential oil is widely used in the food industry as a flavoring, in perfumery, and in the medical industry [17]. Onion (*Allium cepa*) is an essential spice, as well as a commercial vegetable traditionally used for its medicinal properties in the management of various diseases [12,18]. Tomato (*Solanum lycopersicum*) belongs to the Solanaceae family and is one of the most important vegetables used in human nutrition, which has a wide variety of consumption [19]. Currently, tomato production accounts for 25 percent of the total vegetable production in the world [20]. Alfalfa (*Medicago sativa* L.) is also one of the most important fodder legumes on the globe. It is the fourth most widely grown crop in the United States after corn, wheat, and soybeans [21,22]. Thyme (*Thymus vulgaris* L.) belongs to the Lamiaceae family and is distributed in different regions of the Mediterranean and Asia. The volatile phenolic oil of thyme has been reported among the top 10 essential oils to have antibacterial, antifungal, antioxidant, natural food preservative, and antiaging properties in mammals [23]. The introduced crops are common hosts of the *Cuscuta* genus and were used as hosts in this study.

Major studies in relation to the dodder have been done more on its relationship with the hosts as a parasitic plant. In this study, our aim was to evaluate the effect of host species on the health-promoting compounds in field dodder seeds. We tested the hypothesis that more bioactive compounds will accumulate in the seeds if the medicinal plants are hosts for dodder parasites. We did a hypothesis assay by analyzing flavonoid compounds, phytosterols, antioxidant capacity, etc.

## 2. Materials and Methods

*Plant Material and Seed Collections*

This experiment was conducted in an experimental greenhouse in the research department of Parsian Gharch Company, Nazarabad City located in Alborz Province, Iran. The amount of light inside the greenhouse was 1900 + 50 μmol m$^{-2}$ s$^{-1}$. Dodder seeds registered in the seed bank of the University of Zabul (UOZ) with the identification numbers

2432/88 were used in this study. The host plants were planted in polyethylene pots the size of 19.5 × 18 × 14 cm, which were filled with a mixture of leaf soil, compost, and perlite. The host plants were alfalfa (*Medicago sativa* cv. Hamedani), thyme (*Thymus vulgaris* 'Varico 3'), basil (*Ocimum basilicum* 'Persian'), tomato (Hybrid Manaco F1), and onion (*Allium cepa* 'Saryna'), and 40 days after seed germination, 50 seeds of *Cuscuta campestris* were sowed at a distance of 1 to 2 cm around them. After about 40 to 50 days, parasitic plants produced seeds on different hosts, and these seeds were collected in the fully matured stage, then they were sun-dried for 4 days, and the seeds were kept in suitable temperature and humidity conditions (20 °C and 12% relative humidity) for subsequent analyses.

## 3. Materials

All HPLC grade standards, hydrochloric acid, gallic acid, apigenin, phenol, DPPH (2, 2-diphenyl-1-picrylhydrazyl), and Folin–Ciocalteu's reagent were provided from Sigma Aldrich Co. (St. Louis, MO, USA).

### 3.1. Weight of a Thousand Seeds

The weight of 1000 fully matured seeds of the dodder parasite grown on the different hosts was determined after 4 days of exposure to the sun with a sensitive scale of 0.0001 g (AND GR202, Japan).

### 3.2. Determination of Carotenoids, Total Phenol, and Total Flavonoids

To measure the amount of carotenoid, 1 g of seeds was ground in a mortar and pestle, and 20 mL of 80% acetone was added to it; then, the homogenate was centrifuged at $3000 \times g$ for 15 min. Next, 1 mL of the supernatant was taken, and the absorbance was read at wavelengths of 480, 663, and 645 nm, and the amount of carotenoid was determined spectrophotometrically following Metzner et al.'s [24] equation:

$$\text{Carotenoids} = (\text{O.D } 480) + (\text{O.D } 663) \, (0.114) - (0.638) \, (\text{O.D } 645)$$

O.D: The optical density of the seed extract at the noticed wavelength.

Total phenolics were measured based on the colorimetric method using the Folin–Ciocalteu reagent [25]. First, the amount of 0.2 g of the dodder seeds was homogenized in 3 mL of 80% methanol and incubated in a hot water bath at a temperature of 70 °C for 180 min. Then, it was centrifuged at $9000 \times g$ for 15 min, and the resulting supernatant was collected to measure total phenol. Then, 100 μL of the Folin–Ciocalteu reagent was added to 20 μL of methanolic obtained extract. After 5 min, 300 μL of 7.5% sodium carbonate solution was added to the resulting pellets and kept for 90 min in the darkness at 25 °C. Finally, the absorption of each sample was read at 765 nm wavelength by a UV–Visible spectrophotometer, and the content of total phenolics was expressed as mg of gallic acid per g dry matter [25].

To measure flavonoids, ethanol and hydrochloric acid were added to the ground seeds in a ratio of 99:1. The samples were exposed to ultrasonic waves with a frequency of 40 kHz for 30 min. Then, the samples were kept in a hot water bath at 80 °C for 10 min. After cooling, the samples were centrifuged at $6000 \times g$ for 5 min. The absorbance of the samples was determined spectrophotometrically at three wavelengths of 270, 300, and 330 nm, and the amount of total flavonoid was determined using the standard curve of apigenin [26].

### 3.3. Determination of Total Polysaccharides

The total polysaccharide content of the seed samples was quantified by the phenol-sulfuric acid reaction based on Dubois et al.'s [27] procedure using D-glucose as a standard so that 10 mL of 80% ethanol was added to 0.1 g of powdered seeds, and after 24 h, 0.5 mL of the supernatant was collected and made up to 2 mL with distilled water. Then, 1 mL of 5% phenol was added to it and mixed completely. Next, 5 mL of concentrated sulfuric acid was added to it, about 30 min after cooling, and its absorbance was determined by a

spectrophotometer at a wavelength of 485 nm. The standard curve prepared from glucose was used to determine the total polysaccharides.

### 3.4. Galactitol Assay

One gram of the seeds of the dodder was thoroughly ground with liquid nitrogen in a mortar and pestle. Then, 10 mL of $C_4H_{10}O$ was added to the homogenate. The homogenate was placed in an ultrasonic bath in the dark for 25 min. The $C_4H_{10}O$ phase was separated, and its pigments were separated by a 40% ammonia solution. The ammonia phase was discarded, and the $C_4H_{10}O$ phase was removed by a vacuum pump. One microliter of the supernatant was dissolved in methanol for injection into HPLC. Polyols quantification (galactitol) was carried out by a high-performance liquid chromatograph equipped with a UV–Visible (UV–Vis) absorbance detector, a variable loop injector, and a TSKgel ODS−80Tm HPLC column having a 5 µm particle size. One microliter of seed extract was injected into the chromatograph and eluted with acetonitrile: ethanol: $H_2O$ (5:2:3) in the isocratic mode at a 1 mL min$^{-1}$ flow rate. Polyol (galactitol) was measured based on the peak height (Figure 1) by the use of standards and retention time, and calibration curves [28].

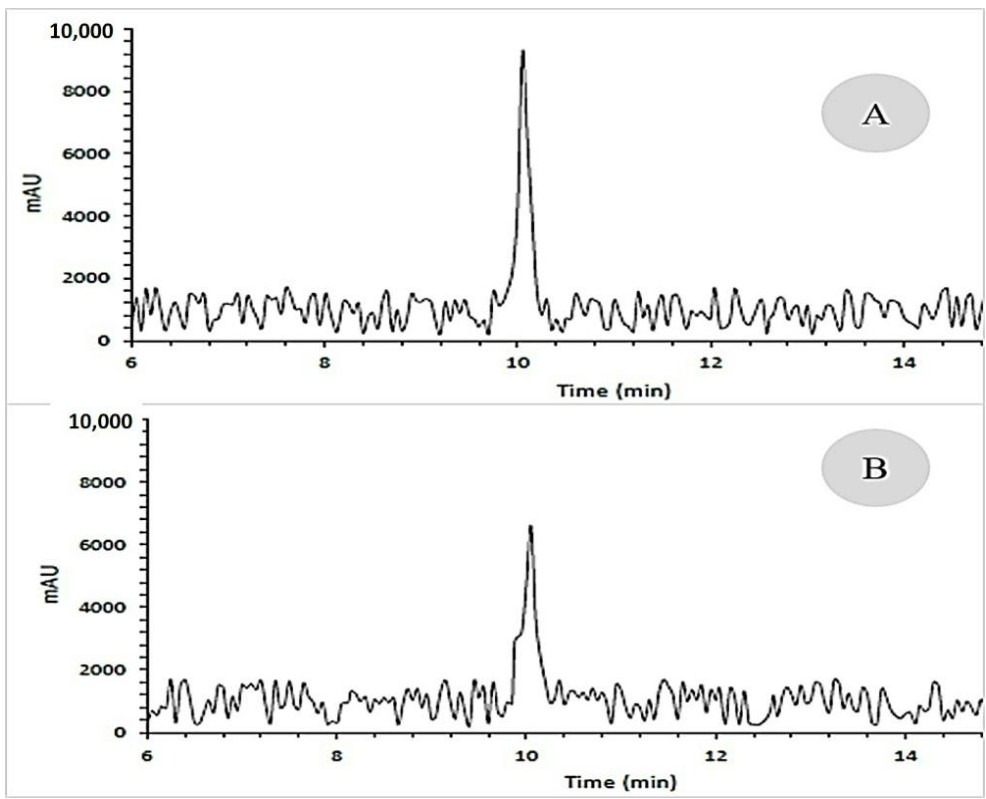

**Figure 1.** HPLC representative chromatogram of galactitol (retention time: 10.05 min): (**A**) standard extracts and (**B**) seed extracts of *Cuscuta campestris* developed on various hosts.

### 3.5. DPPH Radical Scavenging

To determine the DPPH (2, 2-diphenyl-1-picrylhydrazyl), radical scavenging with modified Abe et al.'s [29] method was used. Then, 100 mg of powdered dodder seeds were homogenized in liquid nitrogen and 90% ethanol and kept at 4 °C for 24 h. The insoluble solids were separated using a centrifuge at 3500× *g* for 5 min. Then, 30 µL of the resulting solution was mixed with 800 µL of DPPH dissolved in ethanol (0.5 mM), and finally, the absorbance at 517 nm was determined after 30 min of incubation in the dark. The following equation was used to quantify the DPPH radical activity inhibition capacity: radical scavenging activity (%) = [(Ao − As)/Ao] × 100, where Ao is the absorbance of the control blank, and As is the absorbance of the sample.

### 3.6. Flavonoid Compositions (Quercetin, Bergenin, and Kaempferol)

Hajimehdipoor et al.'s [30] method was used to quantify the content of quercetin and kaempferol. In summary, completely powdered seed samples (1 g) were dissolved in 80% methanol (25 mL) and incubated in an ultrasonic bath for extraction for 60 s. Next, the centrifugation of extracts was done at $3000 \times g$ for 15 min, and the supernatant was collected. The extraction process was performed two more times on the centrifuged residue using 25 mL of 80% methanol. The filtration of the resulting solution was done through a membrane filter (0.40 μm) before injection. After filtration, the supernatant was collected in a 100 mL flask, followed by dilution with 80% methanol. HPLC analysis was carried out using a Waters Alliance device equipped with a vacuum degasser and quaternary detector. The column was ACE C18 (4.6 × 250 mm, 5 μm). The mobile phase was a linear gradient with orthophosphoric acid and acetonitrile at a 1 mL min$^{-1}$ flow rate. The injection volume for all extracts was 10 μL. The levels of quercetin, bergenin (cuscutin), and kaempferol in each sample were quantified from the integrated chromatographic peak height based on the standards and retention time (Figure 2) and calibration curves.

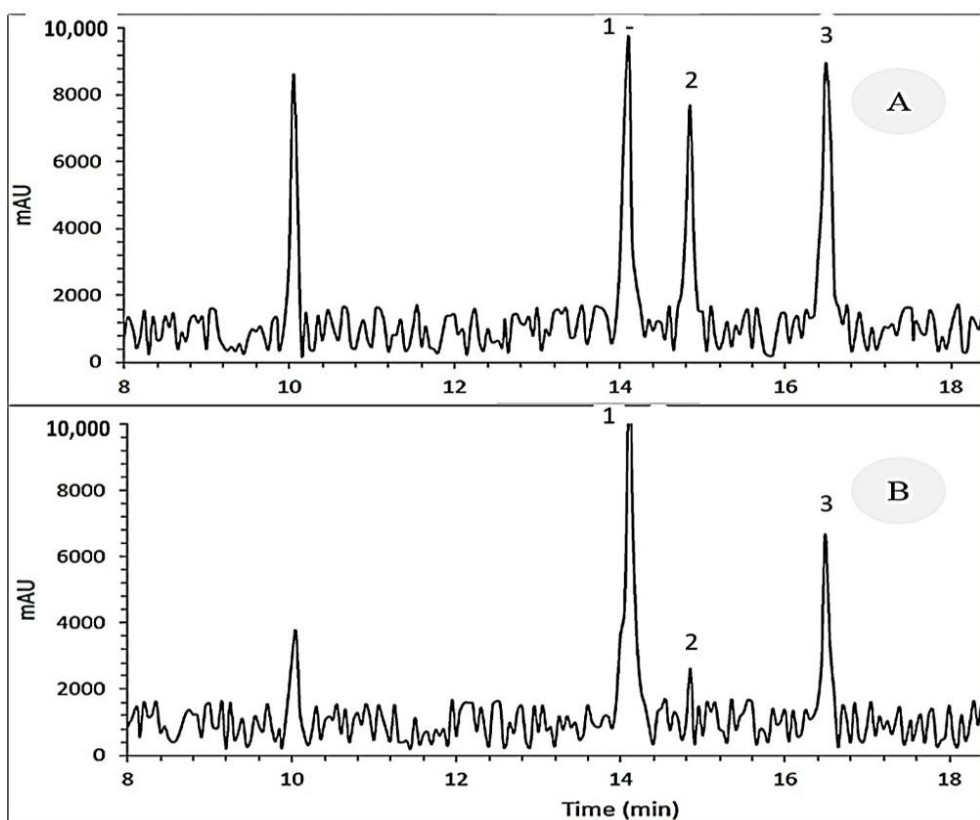

**Figure 2.** HPLC representative chromatogram of (**A**) the standard extracts and (**B**) seed extracts of *Cuscuta campestris* developed on various hosts. 1. Quercetin (retention time: 4.11 min); 2. Bergenin (retention time: 14.85 min); 3. Kaempferol (retention time: 16.49 min).

### 3.7. Analysis of Phytosterols

To measure the contents of phytosterols, the method described by Kamgang Nzekoue et al. [31] was used. One gram of completely ground seeds (100% purity) was incubated in a 50 mL microtube with 1 mL of hydrogen chloride (1N) and 3 mL of water. The resulting solution was subjected to ultrasonic waves for 10 min (60 Hz). After ultrasonication, the extracts were saponified with 5 mL of KOH (50% *w/w*) and 20 mL of ethanol in a water bath at 80 °C for 40 min. After cooling, the phytosterols were washed three times with hexane (10 mL × 3); next, the samples were collected and desiccated using a rotary evaporator. Subsequently, the desiccated extracts were reformed with 1 mL of hexane. Derivatization

of elicited phytosterols for the HPLC assay was conducted via densylation, as described by Nzekoue et al. [32]. Finally, the samples were sonicated and filtered for the HPLC assay.

Danylated phytosterols were determined using high-pressure liquid chromatography (HPLC) equipped with an autosampler, a quaternary pump, and a diode array detector (DAD). The injection volume was 20 μL, and the separation of analytes was done on a Gemini C18 analytical column (250 × 3.0 mm, 5 μm) before a security guard column C18 (4 × 3 mm, 5 μm). The mobile phase was HPLC grade methanol (100%) at a 0.5 mL min$^{-1}$ flow rate. Washing was done in isocratic mode, and phytosterols were quantified and determined at 254 nm. Based on the integrated chromatographic peak height (Figure 3) regarding standards, retention time, and calibration curves, phytosterols such as Brassicasterol, Campesterol, Stigmasterol, β-sitosterol, and Δ$^7$-Avenasterol were detected.

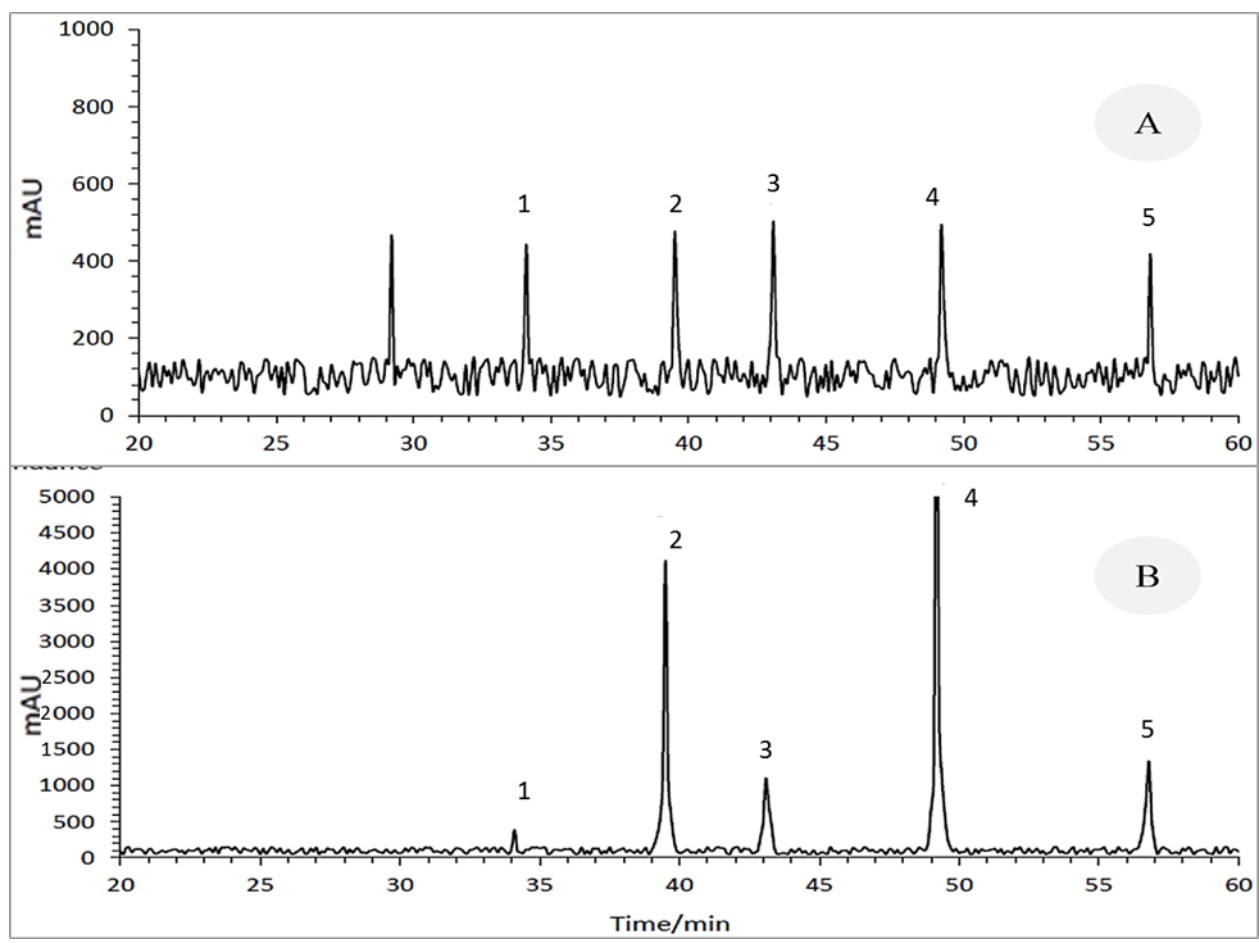

**Figure 3.** HPLC representative chromatogram of (**A**) the standard extracts and (**B**) seed extracts of *Cuscuta campestris* developed on various hosts. 1. Brassicasterol (retention time: 34.1 min); 2. Campesterol (retention time: 39.5 min); 3. Stigmasterol (retention time: 43.09 min); 4. β-Sitosterol (retention time: 49.2 min); 5. Δ$^7$-Avenasterol (retention time: 56.8 min).

### 3.8. Statistical Analysis

Plant cultivation was done with three independent replications in a completely randomized design. The average data obtained from three independent repetitions were subjected to one-way ANOVA analysis by the LSD test and considering the confidence level of $p \leq 0.05$ using SAS software. The significance of the differences was indicated by letters on each column. Different letters indicate significance at the 5% level, and similar and common letters indicate the non-significance of the differences at the 5% level ($p \leq 0.05$).

## 4. Results

### 4.1. The Effect of Host Plant on Seed Weight

The weight of one-thousand seeds developed on different hosts is presented in Figure 4. These results indicate that the effect of various hosts on the weight of one-thousand seeds in this study is not significant (Figure 4).

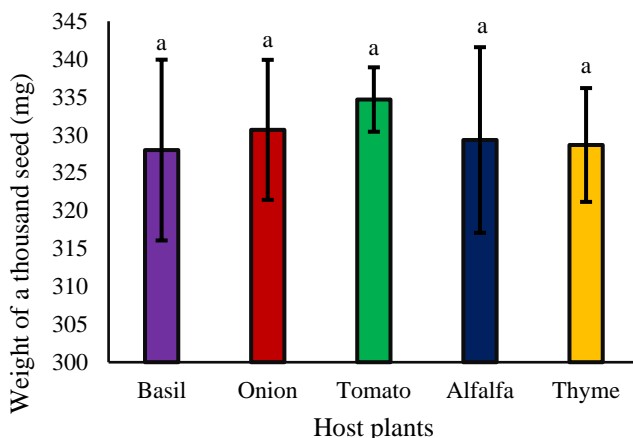

**Figure 4.** The effect of different hosts on the weight of a thousand seeds of field dodder (*Cuscuta campestris*). Bars with the same lowercase letter are not significantly different at $p \leq 0.05$ for $n = 3$.

### 4.2. Galactitol Content, Total Polysaccharides, Carotenoid Accumulation, DPPH Radical Scavenging

In this study, it was hypothesized that the metabolite profile of field dodder (*Cuscuta campestris*) as a holoparasite is affected by the host plant and improves if the host can be a medicinal plant or herbs. Accordingly, this experiment was designed to test this hypothesis with five host plants, including herbs, medicinal plants, fodder, and vegetable crops. Therefore, our approach to the dodder plant was to evaluate the functional compounds of dodder seeds as a medicinal plant, not a common weed.

The accumulation of galactitol in seeds hosted by onions and thyme was higher than the other species; however, the lowest content of galactitol was obtained from seeds hosted by tomatoes (Figure 5A). The total polysaccharide of dodder seeds grown on the different hosts was also noticed. The results revealed that the seeds of dodder plants hosted by onions and thyme had the highest levels of polysaccharides, and the seeds of dodder plants hosted by tomatoes, basil, and alfalfa had the least levels of total polysaccharides (Figure 5B). The results showed that if the host plant is basil or thyme, the total carotenoid content of seed extracts increases; however, if the host plant is alfalfa, tomato, and onion, less carotenoid accumulation occurs (Figure 5C). The antioxidant activity of extracts prepared from seeds of dodder developed on different hosts showed that DPPH radical scavenging was higher in seed extracts developed on thyme and onion hosts, respectively, and the minimum DPPH radical scavenging occurred with extracts of alfalfa-hosted plants (Figure 5D).

The contents of functional flavonoid compounds, including quercetin, bergenin (cuscutin), kaempferol, total flavonoid, and flavonoid compositions in the seeds of dodder grown on the foliage of different hosts, were also studied. Quercetin accumulated more in the dodder seeds collected from the onion host than the other hosts (Figure 6A). In contrast, other hosts had lower quercetin contents than onions (Figure 6A). The concentration of kaempferol was greatest in seeds of plants grown on onion, followed by basil, and was lowest in seeds hosted by alfalfa and tomato (Figure 6B). The kaempferol content of dodder seeds also depends on the type of host plant, so that the maximum amount was obtained under the hosts of onion and basil, respectively, and the least amount was obtained under the hosts of tomato and alfalfa (Figure 6B). The bergenin content was also significantly affected by the type of host, so that the seeds developed on thyme had the highest accumulation of bergenin, and the seeds developed on alfalfa had the lowest content (Figure 6C). The amount of total flavonoid accumulation did significantly respond to the host species.

In the seeds developed on onions and thyme, respectively, the amount of total flavonoids was the greatest. The minimum accumulation of total flavonoids was recorded in seeds developed on tomato, basil, and alfalfa hosts without significant differences (Figure 6D). On the other hand, the maximum content of total phenolics was obtained from seeds developed on thyme, basil, and tomato hosts (Figure 6E). Dodder seeds developed on onions and alfalfa hosts had the least total phenolics (Figure 6E).

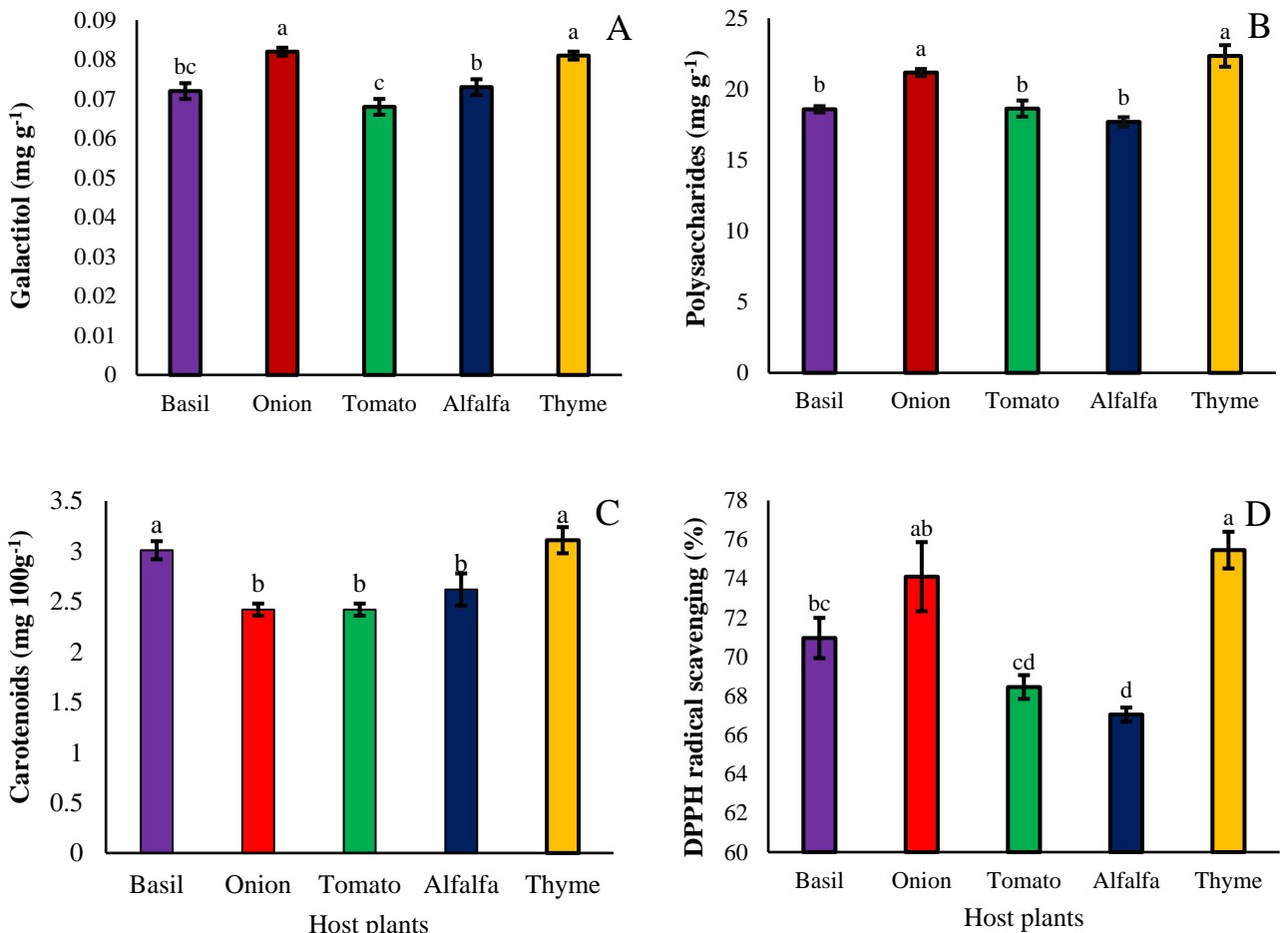

**Figure 5.** The effect of different hosts on the galactitol (**A**), total polysaccharides (**B**), carotenoid content (**C**), and DPPH radical scavenging (**D**) of the *Cuscuta campestris* seed set on different hosts. Bars with the same lowercase letter are not significantly different at $p \leq 0.05$ for $n = 3$.

### 4.3. Phytosterols (Brassicasterol, Campesterol, Stigmasterol, β-Sitosterol, and Δ⁷-Avenasterol)

Plant phytosterols are an important component of membranes in all eukaryotic organelles that are both synthesized by the plant itself and absorbed from the environment, considering that the only source of water and nutrients received by the *Cuscuta* genus is the host plant. The possibility of the effect of the host plants for these parasitic plants on the accumulation and biosynthesis of phytosterols was also assessed in the seeds of the dodder set on different hosts. The brassicasterol content was not significantly affected by the hosts, however; it was higher on basil and thyme hosts, respectively, and lower on tomato hosts (Figure 7A). The campesterol content was significantly affected by the hosts, so that its amount decreased with the alfalfa host and increased in other hosts (Figure 7B). Stigmasterol accumulation was not significantly affected by any of the hosts (Figure 7C). Onions, basil, and thyme hosts increased β-sitosterol accumulation in seeds of the dodder parasite; however, it was reduced with tomato hosts (Figure 7D). Δ⁷-Avenasterol, on the other hand, increased in dodder seeds hosted with thyme, followed by basil, and decreased with onion

(Figure 7E). In general, the accumulation of phytosterols in dodder seeds increased with basil, thyme, and onion hosts and decreased with alfalfa and tomato hosts (Figure 7F).

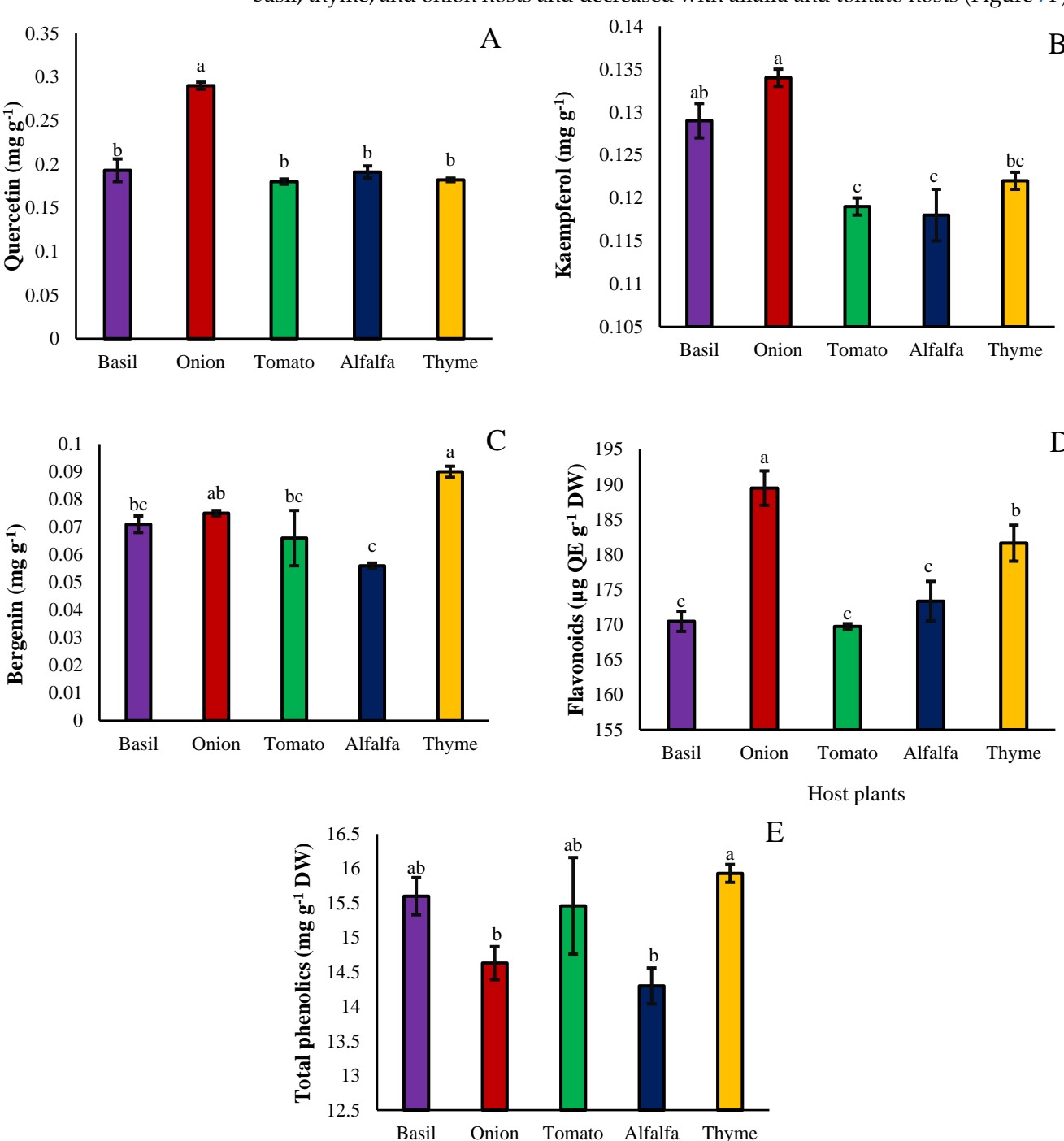

**Figure 6.** The effect of different hosts on quercetin (**A**), kaempferol (**B**), bergenin (**C**), total flavonoids (**D**), and total phenolics (**E**) of the *Cuscuta campestris* seeds set on different hosts. Bars with the same lowercase letter are not significantly different at $p \leq 0.05$ for $n = 3$.

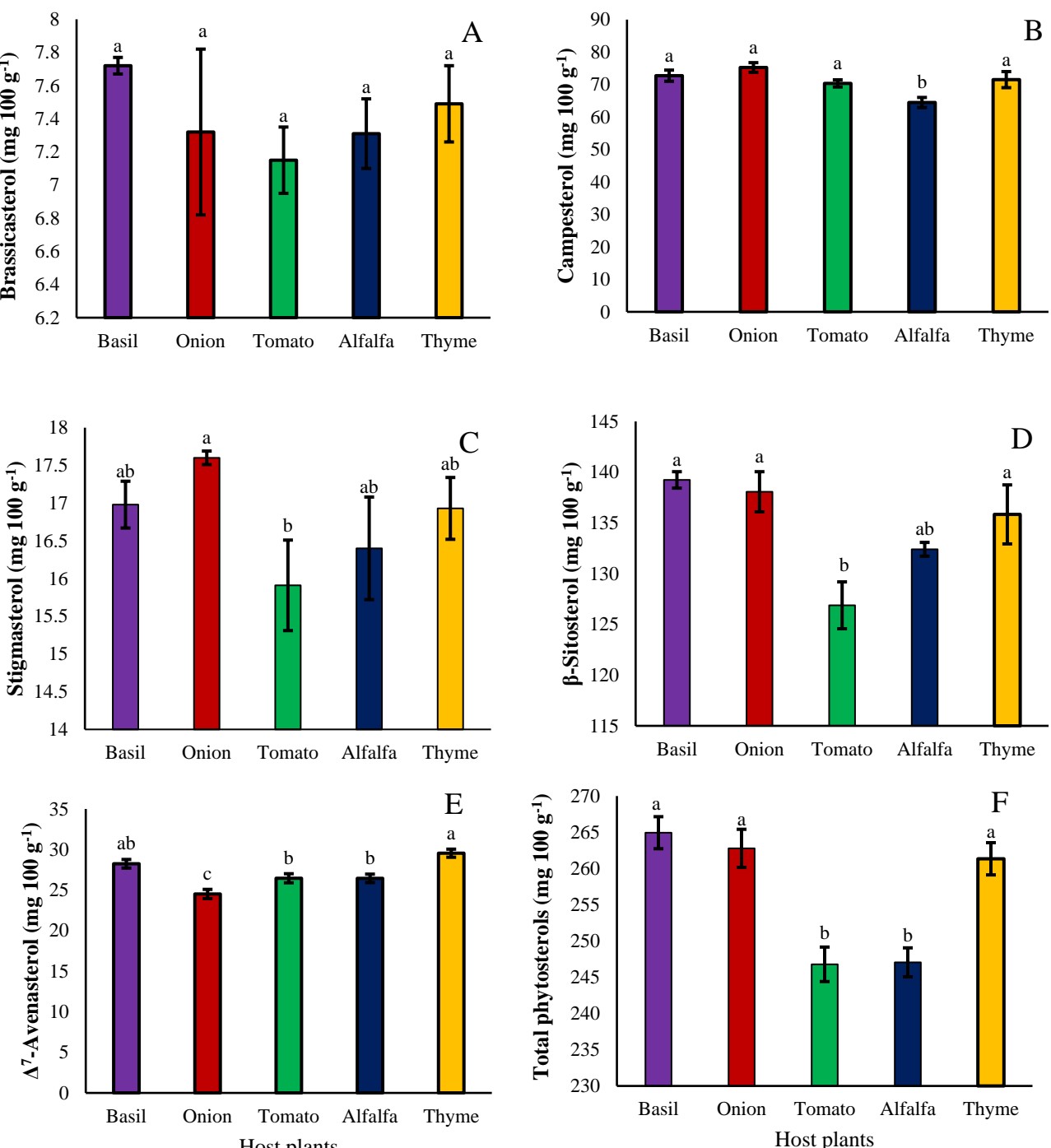

**Figure 7.** The effect of different hosts on brassicasterol (**A**), campesterol (**B**), stigmasterol (**C**), *β*-Sitosterol (**D**), $\Delta^7$-Avenasterol (**E**), and total phytosterols (**F**) of the *Cuscuta campestris* seeds set on different hosts. Bars with the same lowercase letter are not significantly different at $p \leq 0.05$ for $n = 3$.

## 5. Discussion

The main aim of this study was to assess the important secondary metabolites of dodder seeds obtained on five different hosts. We hypothesized that more bioactive compounds will be accumulated in the seeds of the dodder if the hosts of the dodder are medicinal plants. This hypothesis exists in the ethnobotany of different nations, but there is little scientific information about its rejection or acceptance. For example, in the Iranian ethnobotany, it is recommended to use the seeds of the dodder with a host of thyme for the treatment of diseases. Considering that, in addition to the import of water, photosynthetic

materials, minerals, and phytohormones from the host plant by the dodder [3], it has recently been shown that some RNA molecules are also transferred between the host and the dodder in a reciprocal manner that indicates the involvement of a number of genes [4]. These findings indicate the importance of the host species on the metabolism of the holoparasite plants, which may also affect secondary metabolism. To test the hypothesis, basil (herb), onion (vegetable and spice), tomato (vegetable), alfalfa (fodder), and thyme (medicinal plant) were used as host plants.

The results of this research showed that the weight of 1000 seeds was not affected by the species of hosts, but the accumulation of secondary metabolites in seeds was significantly different. The content of carotenoids in seeds hosted by basil and thyme was higher than other hosts. The content of galactitol (dulcitol), which is a polyol, was higher in the seeds hosted by onion and thyme. Seeds hosted by onion and thyme also had higher polysaccharide contents. The results of Kumar and Amir [11] showed that, in addition to photosynthetic substances, the dodder parasite inter many primary metabolites into its organs, which accumulate or catabolizes them into other metabolites. Increasing the level of polysaccharides and galactitol is in line with increasing the antioxidant capacity (DPPH radical scavenging) of the dodder seed extract with thyme and onion hosts.

The levels of flavonoid compounds also showed significant differences in relation to the type of hosts. Quercetin levels were higher in onion-hosted dodder seeds than other hosts. The levels of kaempferol showed the greatest amount under onion and basil hosts, respectively, and the bergenin was higher under thyme and basil hosts, respectively. The highest content of total flavonoid compounds in dodder seeds was found under the onion host, followed by thyme. Considering that flavonoid compounds have a high antioxidant capacity, it seems that the improvement of the DPPH free radical scavenging under the hosts of thyme and onion, followed by basil, is correlated with the increase in the level of total flavonoid in the dodder seeds obtained from these hosts. In this study, the content of phenolic substances was also assayed in the dodder seeds set on various hosts. Thyme, followed by basil and tomato, were better hosts for the dodder in terms of increasing the total phenol content. There are few studies on the biosynthesis of metabolites in different organs of parasitic plants. In some studies, it has been suggested that *Cuscuta campestris* and *Cuscuta reflexa* have no possibility of preferential absorption of certain compounds from the host [33,34]. However, Bais et al. [35] revealed that the biochemical content of the *Cuscuta reflexa* developed on the *Cassia fistula* host is different from that developed on the *Ficus benghalensis* host, and this is dependent on the type of host. Based on this, we suggest that the tissue of different hosts be analyzed and compared in terms of different metabolites. By knowing the level of metabolites in the biomass of hosts and comparing it with the amount of accumulation of metabolites in the organs of the host, it is possible to achieve a kind of marker to predict the accumulation of metabolites in the mutual relationship between parasite and host.

Phytosterols are plant substances with chemical structures and biological functions close to cholesterol, and the major plant sources of these compounds are vegetable oils (particularly unrefined oils), nuts, seeds, and seeds [36]. The most common plant sterols are sitosterol, campesterol, and stigmasterol [37]. Increasing the level of these metabolites in the diet leads to a reduction in cholesterol absorption [38]. Therefore, finding food sources rich in phytosterols can help improve the health of people with high-cholesterol diets. The results of this study showed that the contents of campesterol, stigmasterol, and $\Delta^7$-Avenasterol of dodder seeds were substantially affected by the type of host, but the profiles of brassicasterol and $\beta$-sitosterol were not substantially affected in association with the species of studied hosts. Overall, the content of phytosterols in the seeds of the dodder was higher with basil, onion and thyme hosts, and the lowest content of these compounds was obtained in the seeds with tomatoes and alfalfa hosts. Phytosterols have been previously reported in the seeds of *C. reflexa* [39] and *C. australis* [40] species. The anticancer and tumor-suppressing properties of phytosterols such as campesterol and stigmasterol have been well documented in various studies [41,42]. In a study, Kornsteiner-Krenn et al. [43]

evaluated the amount of phytosterols in 10 types of nuts as one of the most important sources of phytosterols in the diet. The average total phytosterol content ranged from 71.7 mg (Brazilian nut) to 271.9 mg (pistachio) per 100 g of oil. In this study, the levels of studied phytosterols in C. campestris seeds were high in a host-dependent manner. By comparing the results of this study with the results of Kornsteiner-Krenn et al. [43], we found that the dodder seeds developed on basil, onion, and thyme hosts have more phytosterol content than all types of nuts. As a result, they are an excellent source of phytosterols that can be considered for food and pharmaceutical purposes.

These results, in addition to proving the hypothesis of the study that if medicinal plants such as thyme are hosts to the dodder parasite, more bioactive substances accumulate in the organs of the dodder, show that there are other hosts that should be explored in order to improve the level of secondary metabolites in the dodder. In this study, the seeds of the dodder parasite set on onion and basil had appropriate levels of different bioactive substances compared to the seeds set on the thyme host. Taken together, these three hosts are recommended for dodder to use its seeds as a rich source of healthful compounds.

## 6. Conclusions

The results of this experiment showed that the extract of the seeds of the dodder is rich in flavonoid compounds and has high antioxidant activity, as long as the seeds of the dodder are set on thyme and onion. Additionally, in addition to thyme and onion, basil plants are also a suitable host in terms of the accumulation of phytosterols. These findings reveal that if the hosts of the dodder are medicinal plants and herbs, more health-promoting substances are accumulated in the tissues of the dodder parasite. For further studies, we suggest that the tissue of different hosts be analyzed and compared in terms of different metabolites. By knowing the level of metabolites in the host biomass and comparing it with the amount of accumulation of metabolites in the organs of the parasite, it is possible to achieve a kind of marker to predict the accumulation of metabolites in the relationship between parasite and host.

**Author Contributions:** Conceptualization, Y.F.; methodology and resources, D.R.; software, data curation, visualization, and formal analysis, D.R. and Y.F.; validation, M.Z., G.S., G.K. and A.B.; investigation, M.Z., G.S. and G.K.; writing—original draft preparation, Y.F. and M.Z.; writing—review and editing, Y.F., M.Z., G.S., G.K. and A.B.; and supervision, project administration, and funding acquisition, D.R. and M.Z. All authors have read and agreed to the published version of the manuscript.

**Funding:** This research received no external funding.

**Institutional Review Board Statement:** Not applicable for studies not involving humans or animals.

**Data Availability Statement:** Not applicable.

**Acknowledgments:** This work was supported by RUDN University Strategic Academic Leadership Program.

**Conflicts of Interest:** The authors declare no conflict of interest.

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
