# Peer review of "An Insight into Cuscuta campestris as a Medicinal Plant: Phytochemical Variation of Cuscuta campestris with Various Host Plants"

_agriculture, doi:10.3390/agriculture13040770_

Round 1

Reviewer 1 Report

1.      Because there are no real interaction issues in this study, “Interaction” in the title is inappropriate. “ change” or “alternation” may be better.

2.      For all figures, there are no error bars for each group (n=3), why?

3.      Please explain what are the “standard extracts”.

4.      For the biomass, the study only provides the weight of one-thousand seeds, which only investigates the quality of the seeds. I suggest that the total biomass of the seed is also needed to show the effect of the host on seed production.

5.      The compound profiles in this manuscript are quite different from that are published in the literature. I wonder if the seed (identification number 2432/88) really is Cuscuta campestris.

6.      The descriptions in Lines 282-288 are very unclear, please revised.

7.      The description in Line 300 is unclear, please revised.

8.      In Line 374, the lowest is usually only one, please describe clearly.

9.      In line 282, “soecies” is a typo.

Author Response

Agriculture (ISSN 2077-0472)

Dear Reviewer:

We are very grateful to you for the opportunity to revise our manuscript, An Insight into Cuscuta campestris as a Medicinal Plant: Phytochemical Alternation of Cuscuta campestris with Various Host Plants. We appreciate the careful review and beneficial suggestions. It is our opinion that the manuscript is markedly improved after making the suggested edits. Following this letter are the reviewer comments with our responses. Changes made in the manuscript are marked using Word Track Changes. The revision has been developed in consultation with all coauthors, and each author has given approval to the final form of this revision. 

Regarding the findings of this article, we very much hope the revised manuscript is accepted for publication in Agriculture.

Sincerely,

Best Regards

Dariush Ramezan1 and Meisam Zargar2

1 Department of Horticulture and landscaping, Faculty of Agriculture, University of Zabol, Zabol, Iran

2 Department of Agrobiotechnology, Institute of Agriculture, RUDN University, 117198 Moscow, Russia

Email: drhorticul@uoz.ac.ir ; zargar_m@pfur.ru

  • References revised according to the editor comment.
  1. Because there are no real interaction issues in this study, “Interaction” in the title is inappropriate. “ change” or “alternation” may be better.

Authors' comment: title was modified.

  1. For all figures, there are no error bars for each group (n=3), why?

Authors' comment: error bars were added in the text.

  1. Please explain what are the “standard extracts”.

Authors' comment: All HPLC grade standards were provided from Sigma Aldrich Co. (St. Louis, MO, USA). The measurements were made by comparing the retention time of the samples with the standard (Sigma-Aldrich), which is explained in the Materials section (2. Material and methods).

  1. For the biomass, the study only provides the weight of one-thousand seeds, which only investigates the quality of the seeds. I suggest that the total biomass of the seed is also needed to show the effect of the host on seed production.

Authors' comment: Due to the differences in the size and planting density of the host plants, the seed yield was not considered an accurate trait.

  1. The compound profiles in this manuscript are quite different from that are published in the literature. I wonder if the seed (identification number 2432/88) really is Cuscuta campestris.

Authors' comment: In other literature, metabolites have been evaluated in organs such as shoots, haustoria and flowers. The growth climate and the type of hosts also make a big difference in the amount of its metabolites. The seeds of this plant were collected by botanists at the university of Zabol and are kept in the seed bank of the university.

  1. The descriptions in Lines 282-288 are very unclear, please revised.

Authors' comment: It was modified.

  1. The description in Line 300 is unclear, please revised.

Authors' comment: It was modified.

  1. In Line 374, the lowest is usually only one, please describe clearly.

Authors' comment: It was corrected.

  1. In line 282, “soecies” is a typo.

Authors' comment: Corrected

Reviewer 2 Report

Review of the manuscript 2248185

The authors of the manuscript made an attempt to evaluate the effect of host species on the health-promoting compounds found in field dodder seeds. The paper presents quite interesting results. The objective of the study should be reformulated. The title should be more precise. The rest of the manuscript needs some changes as well. English linguistic correction should be made. Therefore I recommend major revision of the manuscript.

More detailed comments are given below:

Line 15 – Cuscuta, not Cascuta

line 29 – one dot should be deleted

line 83 – tomato, not Tomato

line 94-97 – The aim of the study should be reformulated. The authors haven’t studied the bioactive effect of secondary metabolites of dodder seeds. Fragments “pay attention to the effect of the host on chemical profile” and “ to evaluate the effect of host species on the health-promoting compounds found in field dodder seeds” provide the same information, it should be shortened.

Line 105 – seeds of what species have the number 2432/88? Of dodder? It should be precised.

line 107 – is the pot size in cm? The information should be added.

There are Figure 2, 4 and 6 inserted into Materials and methods section. They are present in the text before Figures 1 and 3… Figures should be consecutively numbered.

line 232 – letters indicate significance of differences, not significance of results

line 236 – “The effect of host plant on seed weight” – it seems that a fragment of this sentence is missing

line 248 250 – this phrase should be deleted, it repeats the information given in the materials and methods

line 259 – 262 - the description of the Figure 3 D is not precise enough from the statistical point of view. Please check also the description of other figures, especially when letters indicating significance of differences of means are like “ab”, “bc” and not a, b, c.

Standard error bars should be added to figures 1, 3, 5.

References to figures should be deleted in the discussion section.

Author Response

Dear Reviewer:

We are very grateful to you for the opportunity to revise our manuscript, An Insight into Cuscuta campestris as a Medicinal Plant: Phytochemical Alternation of Cuscuta campestris with Various Host Plants (agriculture-2248185). We appreciate the careful review and beneficial suggestions. It is our opinion that the manuscript is markedly improved after making the suggested edits. Following this letter are the reviewer comments with our responses. Changes made in the manuscript are marked using Word Track Changes.  consideration.

Sincerely,

Best Regards

Dariush Ramezan1 and Meisam Zargar2

1 Department of Horticulture and landscaping, Faculty of Agriculture, University of Zabol, Zabol, Iran

2 Department of Agrobiotechnology, Institute of Agriculture, RUDN University, 117198 Moscow, Russia

Email: drhorticul@uoz.ac.ir ; zargar_m@pfur.ru

Line 15 – Cuscuta, not Cascuta

Authors' comment: Corrected

line 29 – one dot should be deleted

Authors' comment: Corrected

line 83 – tomato, not Tomato

Authors' comment: Corrected

line 94-97 – The aim of the study should be reformulated. The authors haven’t studied the bioactive effect of secondary metabolites of dodder seeds. Fragments “pay attention to the effect of the host on chemical profile” and “ to evaluate the effect of host species on the health-promoting compounds found in field dodder seeds” provide the same information, it should be shortened.

Authors' comment: the purpose of the present manuscript was modified to be shortened.

Line 105 – seeds of what species have the number 2432/88? Of dodder? It should be precised

 Authors' comment: This identification number is related to dodder seeds, which is now more clearly stated in the text.

line 107 – is the pot size in cm? The information should be added.

Authors' comment: the size of the pot is in cm.

There are Figure 2, 4 and 6 inserted into Materials and methods section. They are present in the text before Figures 1 and 3… Figures should be consecutively numbered.

Authors' comment: According to this suggestion, figures are consecutively numbered.

line 232 – letters indicate significance of differences, not significance of results

      Authors' comment: Corrected

line 236 – “The effect of host plant on seed weight” – it seems that a fragment of this sentence is missing

Authors' comment: It was revised.

line 248 250 – this phrase should be deleted; it repeats the information given in the materials and methods

Authors' comment: It was revised.

line 259 – 262 - the description of Figure 3 D is not precise enough from the statistical point of view.  Please check also the description of other figures, especially when letters indicating significance of differences of means are like “ab”, “bc” and not a, b, c.

Authors' comment: Descriptions modified

Standard error bars should be added to figures 1, 3, 5.

Authors' comment: It was added.

References to figures should be deleted in the discussion section.

  Authors' comment: It was deleted.

Round 2

Reviewer 1 Report

In the article, there are many "the highest" and "the lowest", however, they should indicate always only one,  it can not be two or three. Please describe clearly.  

Author Response

Dear reviewer 

Your comment addressed in the text. 

Regards

Reviewer 2 Report

The manuscript has been substantially improved by the authors. Most corrections has been made. I have only one remark on the misunderstanding concerning description of results. The authors has changed figure captions, however my intention was to change description of figures in the text (in the section „results”). Figure captions were better in the first version of the manuscript.

I’m explaining on the example of Figure 5D:

The authors write: „The antioxidant activity of extracts prepared from seeds of dodder developed on different hosts showed that DPPH radical scavenging was higher in seed extracts developed on thyme and onion hosts, respectively, and the lowest DPPH radical scavenging occurred with extracts of alfalfa-hosted plants (Figure5D).” The description of the Figure 5D is not precise enough from the statistical point of view. DPPH radical scavenging in alfalfa-hosted plants was significantly lower than in basil, onion and thyme but difference between alfalfa and tomato-hosted plants wasn’t significant.

Please check and correct the description of results in the text (my remark concerns Figures 5, 6 and 7 where differences are not as clear to be marked by single letters a, b or c).

Author Response

Dear Reviewer

We gratefully acknowledge the detailed revision of the text and useful suggestions to improve the paper by the you. We have closely followed your suggestions and introduced the required changes in the text. Main changes are performed in YELLOW.

We hope that after these enhancements the manuscript can now be accepted for publication, although we are certainly willing to consider further changes if necessary.

Yours sincerely,